# An Examination of Classical Art Impact and Popularity through Social Media Emotion Analysis of Art Memes and Museum Posts

**Sofia Vlachou and Michail Panagopoulos \***

Department of Audio & Visual Arts, Ionian University, 49 100 Corfu, Greece
* Correspondence: mpanagop@ionio.gr

**Abstract:** On Instagram, we have all seen memes. Honestly, what would you do if you encountered a meme in a museum? The purpose of the study is to evaluate the nexus between posts uploaded by museum visitors and emotions, as well as the popularity of artworks and memes. We gathered $N = 4.526$ ($N = 1.222$ for memes and $N = 3.304$ for museum posts) entire posts using API. We selected the total number of likes, comments, frequency, nwords, and text emotions as indicators for several supervised machine learning tasks. Moreover, we used a ranking algorithm to measure meme and artwork popularity. Our experiments revealed the most prevalent emotions in both the memes dataset and museum posts dataset. The ranking task showed the most popular meme and museum post, respectively, that can influence the aesthetic experience and its popularity. This study provided further insight into the social media sphere that has had a significant effect on the aesthetic experience of museums and artwork's popularity. As a final point, we anticipate that our outcomes will serve as a springboard for future studies in social media, art, and cultural analytics.

**Keywords:** memes; museums; aesthetic experience; art; emotion analysis; machine learning; ranking; popularity; social media

## 1. Introduction

The pervasive usage of the Internet and the rapid absorption of information about global events have generated a new visual culture known as meme culture [1]. Based on their contents, memes are classified into many groups. They employ comedy to address numerous topics, including politics, the economy, work–life, and personal relationships [2]. The pandemic crisis of COVID-19 has altered social media usage in all facets of employment, accurate information, and daily activities [3]. Due to its eye-candy design [4], Instagram has grown rapidly in recent years compared with other platforms [5]. During the lockdown, an upsurge in Instagram posts in general was observed, as well as artistically inclined memes [6]. Instagram is comparable to Kinfolk magazine in respect to its poetic design. Its minimalist creative aesthetics are adopted by an increasing number of influencers globally [7]. On the other hand, there has been much conversation about how to enhance visitors' art experiences. Varutti [8] agreed that, with proper design, all museums can elicit a wide range of emotions, which can affect viewers' cognitive and emotional well-being. Visitors may, in fact, share their aesthetic experiences on social media. Many museums were temporarily shuttered as a mandatory part of government restrictions imposed around the world in response to the COVID-19 pandemic. As a side effect, quite a lot of them have shared their masterpieces through their Instagram accounts. They took a risk by challenging themselves to keep their audience engaged in art while simultaneously presenting a variety of intriguing stay-at-home activities for them to participate in [9]. Thus, museums should "choose to inspire rather than manipulate in order to motivate people" [10]. During the lockdown period, Instagram claimed that 500 million people were using stories per day, outpacing other competing platforms such as Snapchat [11]. The digital culture and new

media guru Lev Manovich proposes three distinct museum qualities in his presentation titled "Museums and Metaverse" [12]: the ambiance and aesthetic of their venues, the museum's holdings, and the special status of art. He stresses the critical importance of these features in promoting museums in physical or digital hybrid spaces. Furthermore, the "Empathetic Museum" [13] group is noteworthy since it is comprised of museum professionals, with the mission of optimizing "institutional empathy". The goal is to create a strong bond between the museum and society. They intend to achieve this by implementing a well-balanced mix of best practices in exhibition design and programming, community engagement, staff management, and social media marketing and engagement.

Additionally, the majority of Instagram influencers use the platform to promote their self-brand and the idea of the perfect life, microcelebrity, or nostalgia [14]. Outfits, makeup tutorials, parties, coffee breaks, and artworks are all featured on their social media profiles. A quick search on social media reveals that many memes are based on classical paintings [2]. Then, in an attempt to provide a broader perspective on our topic, we provide concrete examples. When most people visit a museum, they are looking for the most captivating artworks. In 2020, the Hermitage Museum in Russia fulfilled an intriguing project titled "hashtag #MUSEUM". The research examined Instagram posts to find out what visitors noticed the most in the museum, which exhibits and spaces were the most photographed, and how they moved around the space. The main categories were selfies, people, and spatial photographs, but images of museum objects were the most popular category [15]. In other words, a visitor to the Louvre will seek out the most well-known pieces of art, such as Leonardo da Vinci's Mona Lisa (Gioconda), Michelangelo's Dying Slave, and Alexandros of Antioch's Venus of Milos. Visitors to Italy's Pinacoteca di Brera will see Francesco Hayez's famous painting The kiss (Il bacio), while visitors to Greece's Acropolis Museum will see the Caryatids or other significant sculptures from Greek history and culture such as the Parthenon Frieze. Therefore, we examine the link between museum posts and aesthetic experience, as well as the popularity of classical art paintings and memes. This study examined the concept of "meme" in the context of art and museum popularity. The remainder of the paper is structured as follows: Section 1 provides an introduction, as well as a description of the purpose and research questions. Section 2 investigates the related work to the topic. The methodology of the study is discussed in Section 3. Section 4 is dedicated to the implementation and includes the experimental results. Section 5 describes the study's limitations. Section 6 is the discussion. Finally, Section 7 concludes the paper and suggests further research.

*Aim and Research Questions*

Viewing art is a dynamic, emotionally arousing, and engaging experience [16]. As a consequence of the massive usage of social media, the art experience has shifted from physical museum settings to the digital arena [4]. Previous research, such as that conducted by Leaver, Highfield, and Abidin [14], has emphasized the Instagram philosophy, which encompasses commercial, creative, and regular use, as well as economic and cultural acceptance. A quick Instagram scroll offers posts regarding news [17], fashion [18], lifestyle [19], self-image presentation [20], and of course arts [21,22]. Abidin [23] points out that Instagram (a) promotes itself as "networked intimacy" by referring to its users as friends, not as followers; (b) encourages users to navigate the app using their "mobile phones" for instant access from anywhere; (c) collects users' moments as keepsakes; and (d) captures spontaneous "happenings" as they occur in real time.

These occasions also involve the sharing of artistic memes and their original museum artworks. We found no prior research in this area. Hence, we conducted both classification and ranking tests to investigate the popularity of art memes on museums in the social media atmosphere. The objective of this study is to assess the link between museum posts uploaded by visitors and their emotions, as well as the popularity comparison between artworks and memes by responding to three research questions (RQs) derived from several machine learning tasks. We concentrated on our original idea, which encompasses both

planned and unplanned visits by viewers to see the artworks they might have encountered previously as memes, as well as their interaction with them. The first question is emotion-based and related to the effect of artistic memes and museum posts. The other two deal with popularity measurement using ranking according to the frequency of occurrence of each meme or its museum post, as well as the number of likes, comments, and nwords associated with artistic memes and museum artwork posts, respectively.

1.  Classification Task for both memes and museum posts:
    - Emotion identification: What emotions inspire people to like, comment, and share posts?
    - RQ1: How do likes, comments, frequency, and nwords relate to emotion?
2.  Ranking Task for both memes and museum posts:
    - RQ2: What is the popularity of artistic memes?
    - RQ3: What is the popularity of museum artworks?

The emergence of social media has had a perplexing effect on how we interact with art. The originality of our research is the popularity comparison between the artistic memes shared during the lockdown and the museum visitors' Instagram posts of those memes after the lockdown. Art dissemination on social media has skyrocketed in the wake of the COVID-19 outbreak, which may offer the ideal opportunity to investigate Instagram's online art engagement. Daily, a significant number of influencers upload photographs captured in museums and galleries. They frequently advocate a luxurious and artistic lifestyle. Furthermore, we expect that our analysis will shed light on how artworks may become popular through social networks. It will also provide museologists, social media scientists, art marketers, artists, and researchers a bird's-eye view of the experience and popularity of art. To the best of our knowledge, this is the first study that examines the popularity of memes and museum posts.

Prior to the actual analysis of our hypothesis, we anticipated distinguishing some of the most important terms. As identified in the literature, Classical art, or Classicism, was introduced in the seventeenth century to represent pieces of art that are inspired by ancient Roman or Greek culture, architecture, literature, and art. During the Renaissance, when classicism reached its pinnacle in Western art, mythological themes such as gods and heroes were frequently portrayed in artworks. Classicism is characterized by harmony, simplicity, restrained emotion, structural clarity, and adherence to established norms of form and craftsmanship [24,25]. In broad terms, Influencers engage primarily in self-branding practices. The objective is to maintain an involved and growing audience that supports their dynamic creative content. Typically, they begin as simple users and gradually adopt techniques and strategies to garner high visibility, establish a career, and obtain income. As can be seen, they have thousands of followers, likes, comments, and views. Their accounts vary from the regular user's accounts in that they incorporate professional features such as paid partnership, a contact button, statistics, and more [14]. Finally, there seems to be no general definition of art popularity in the literature; instead, there is mostly the term popular arts. For this reason, we define popular and popularity as formulated in the Merriam-Webster dictionary [26]. Thus, popular "is something which relates to the general public, suitable to the majority, frequently encountered or widely accepted, commonly liked or approved", and popularity is "the quality or taste of being popular". In social science, consumer behavior, and recommendation systems studies, the term popularity has also been deployed. The present research employed the term art popularity to illustrate how popular certain artworks are or might become through social media networks.

## 2. Related Work

### 2.1. History and Virality of Internet Memes

Over the past two decades, the study of memes has received much attention. Some preliminary work was carried out by Richard Dawkins in his book "The Selfish Gene" in the early 1970s. Dawkins fathered the meme from a biological perspective. He defined

memes as "small cultural units of transmission, analogous to genes, which are spread from person to person by coping or imitation [ . . . ] Like genes, memes are defined as replicators that undergo variation, competition, selection and retention" [27]. Various approaches to studying the link between memes and culture have been proposed. Meme culture [28,29] has evolved as a result of the widespread use of the Internet and the rapid assimilation of knowledge about what is occurring across the world [30]. This has led authors such as Blackmore to examine this notorious issue in her most influential book *The Meme Machine* [31].

Through the prism of communication, Limor Shifman [31] illustrates the infectious nature of memes with more recent evidence. In *The Meme Machine* [32] Susan Blackmore uses the term internet meme "indiscriminately to refer to memetic information in any of its many forms; including ideas, the brain structures that instantiate those ideas, the behaviors these brain structures produce, and their versions in books, recipes, maps and written music" [32]. In contrast, Shifman defined the Internet meme more precisely as "(a) a collection of digital items that share common characteristics of content, form, and/or stance; (b) are created with awareness of each other; and (c) are circulated, imitated, and/or transformed via the Internet by a large number of users worldwide" [28]. She also draws attention to dividing the concept between viral content and memetic content. In a nutshell, she argues that viral content is a high-speed and time-consuming process. The content is shared among individuals on social media networks [28]. It is considered "viral content" because of the message it spreads. When similar derivatives are created from this material, it is known as "memetic content" [28].

Davison [33] has mentioned the transient nature of memes. They can be easily converted and communicated [34]. Furthermore, he proposes three changeable aspects of memes: the manifestation of a meme (visible and external phenomena), the behavior of a meme (the individual's attempt to create the meme, such as the photographic technique or software utilized), and the ideal of a meme (the notion or the idea expressed). When a meme is altered, these three elements also change. Not only do their characteristics change from user to user but so does the medium. According to Bauckhage, Kersting, and Hadiji [35], online memes emerge on platforms such as Tumblr, 4chan, and YouTube and gain popularity on social, news, and entertainment platforms such as Reddit, Facebook, Twitter, and Instagram. Instagram was chosen for this study because it is different and more integrated than other social media platforms.

### 2.2. Sharing Museum Art on Instagram

A growing body of literature has analyzed social media usage, but Instagram's mediating role in reshaping the art experience has received scant attention [36,37]. In his study, Suess [38] divided a typical museum visit into three stages: pre-visitation (stimuli to visit a museum), during-visitation (physical presence at the museum), and post-visitation (outcomes, sharing posts), as you see in the next section. Stylianou-Lambert [39] illuminated the motives for photographing in art museums. She decided that the primary purpose of taking photographs in a museum is to preserve the memory of this experience. Other incentives include emotional satisfaction, educational purposes, and leisure. Consequently, users contemplate a potential visit after viewing these posts.

According to Budge [40], Instagram may enhance the art experience and cultural organization involvement. In persuading people to share information online, she highlights the importance of space. She also believes that photography allows museum visitors to define their own area. The phrase "self-place making" was adopted by the author to support this perspective. Moreover, in his book *Instagram and Contemporary Image*, Lev Manovich [7] defines "Instagrammism" as a mix of media form and content, occasionally with akin style, attitude, and tone. Users establish eye-catching Instagram accounts with their coffee, food, selfies, or other subjects [41] to share their narratives or experiences and engage with others without boundaries [7].

Some academics have referred to selfies as an art form [42]. Heydeman [43] and Burness [44] asserted that Vincent van Gogh and Rembrandt's self-portraits are the earliest selfies. Piancatelli, Massi, and Vocino [22] found in a recent study in this field that museum selfies can enhance visitor interaction with artworks. They claim that the procedure is also influenced by the visitor's emotional response to the art encounter. They also categorized the visitors into four groups based on their emotions, engagement, and favorite photographic style: reality escapers, art enthusiasts, photoholics, and selfie lovers. Kozinets, Gretzel, and Dinohpl identify various types of selfies, including mirror selfies, blending into art, iconic selfies, and contemplative selfies [45]. In Chlebus-Grudzien [46], three types of museum photography were defined as (a) selfies, (b) interaction with space, and (c) interaction with heritage. We intend to apply the following aspects of museum posts and memes to our study.

### 2.3. The Illusion of Popularity and Reputation

The concern of whether the art or the artists have the potential to be popular has been preoccupying experts for some time [47–49], but very little is known about that issue. These theories have their roots in a recent study of 90 twentieth-century artists [50] Scholars have discovered that an artist's social network is the most essential factor in determining their success and reputation. As determined by artists' references in twentieth-century literature, artists with a more diversified social network (personal and professional ties from many fields) were statistically more likely to achieve success than those with a more homogeneous network [50]. Alain Quemin [51], a renowned sociologist, undertook a study from a sociological perspective that employed ranking to quantify the reputation of French contemporary art galleries based on several criteria. The results revealed a correlation between the financial health of the galleries and the reputation of their artists. It demonstrates a strong hierarchy among French contemporary art galleries, with certain foreign galleries occupying the highest ranks.

Anybody could wonder "What is a name?", as Cleeremans et al. [52] found in their research. They have shown that the existence of an artist's name can influence the ultimate aesthetic evaluation and popularity of an artwork, particularly among non-expert perceivers. To comprehend the online context, Kang, Chen, and Kang [53] investigate how the relationship between artists and followers influences the popularity of each artwork. Similarly, how influential is the popularity of the most-liked artworks on the artists' creative process? According to the data, these interactions positively affected the number of likes and comments. The most popular artworks are affected by the artists' personal experiences and interactions with their followers, but not by the process of creation. Let us take a glance at how the conceptual perspective described above relates to art as experience.

Moreover, in our most recent research [54], on social media art's popularity, we discovered that a public art installation that provokes strong emotions may gain popularity through social media sharing practices and habits. We used Instagram and Twitter data to present a distinct perspective on how an artistic installation, such as "Arc de Triomphe, Wrapped" by Christo and Jeanne-Claude in 2021, on a world-famous monument, might emotionally touch spectators and raise the artwork's appeal. The total number of likes, comments, and other analyzed features increased dramatically when the monument was wrapped as compared with when it was not. In this instance, we noticed that a combination of inner qualities may provoke intense emotions that are expressed on social media. In our previous study, we predicted emotions before and after the wrapping of a monument, although in this study, we focused on emotion prediction of classical art memes and the corresponding artworks.

### 2.4. The Museum Experience

Note that this study aims to examine the link between museum posts and emotions. Having that in mind, art experience as a psychological state is capable of evoking a wide range of intense emotions over time [55,56]. Cskszentmihályi [57] and Cskszentmihályi

and K. Rathunde [58] contend that "the person experiences extraordinary emotions and even a unique interaction and fluency with the artwork". The assumptions made by Leder et al. [59] in their theoretical model of aesthetic appreciation and aesthetic judgment appear to be well-grounded. The model consists of five stages that describe how information is processed during an aesthetic encounter. In a nutshell, the process consists of perception, implicit memory integration, explicit classification, cognitive mastery, and assessment, with each stage constituting a feedback loop. They portrayed the experience end to end in a masterful approach, taking into account the previous individual's experiences, knowledge, and emotions.

Following a brief explanation of how aesthetic experience is defined and which variables affect the emotional states of perceivers, we examine how Instagram's role as a mediator transforms the art experience.

Since the inception of Instagram in 2010, little study has been conducted on its use in general, and even less in a museum setting. This has prompted authors such as Suess [38] to suggest integrating Instagram into museum visits. Suess's [38] Figure 1 represents a typical museum visit and is divided into three stages: pre-visit, during-visit, and post-visit; (a) the pre-visit stage includes stimuli that compel the individual to visit a museum. Other visitors' posts about their museum experiences are a great source of inspiration, (b) the second stage entails a visit to the museum by physical presence. To enhance the art experience, visitors are urged to take photos, write comments, and share them on social media, and (c) the final stage focuses on the museum visit outcomes. Following their visit, viewers use Instagram to share their interests and emotions [60]. The above theoretical framework is now be applied to analyze real-world data that is pertinent to the study's objectives.

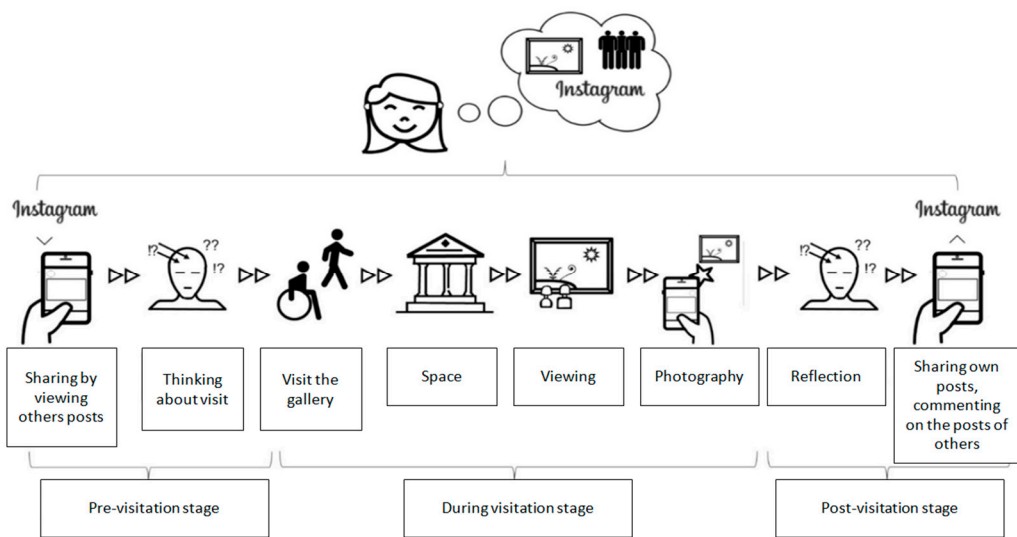

**Figure 1.** Art Gallery Visitor Instagramming [38].

## 2.5. Analyzing Emotions

Primarily in the last decade, there has been little discourse on Instagram emotion analysis. Several publications have appeared documenting sentiment or emotion detection in marketing, COVID-19 pandemic concerns [61], the latest monetary changes [62], and news [63], but not aesthetic experience and online art popularity. Scheibe et al. [64] brought an interesting approach to their study on Miley Cyrus' Instagram account, which received controversial comments owing to her "Wrecking ball" music video. They analyzed the user comments from both the singer's official Instagram account and fan pages. They conducted sentiment analysis using the AFFIN lexicon and POS-Tagger from the NLTK library to identify the polarity of positive and negative comments. They found that a majority of comments were spam. Another study by Aisyah et al. [65] performed Instagram sentiment analysis on politicians' accounts using Naïve Bayes and Random Forest.

They compared the algorithms' accuracy of sentiment analysis with and without sarcasm detection. They achieved a score between 60% and 72 %, correspondingly. Scholars such as Alzamzami et al. [66] draw attention to sentiment classification on short texts using a Domain-Free Sentiment Multimedia Dataset (DFSMD). They used Light Gradient Boosting Machine (LGBM) and six state-of-the-art sentiment classification algorithms to identify core sentiments such as positive, negative, or neutral. Their findings revealed that the LGBM algorithm obtained a substantially higher score. Our research approached the analysis of emotions through the aesthetic experience of the viewer, we also include studies that examined the emotions based on theoretical models and taxonomies of emotions and not merely on sentiments. In their study, Park, Bae, and Cheong [67] employed the NLTK sentiment analyzer for emotion detection as well as an emotion embedding model based on the eight emotions of Plutchik's emotion model. The outcomes of their experiments indicate that their idea was feasible in sentence-level analysis. In a similar way, Asghar et al. [68] carried out sentence-level emotion detection using rule-based classification in order to improve the performance of emotion-based sentiment analysis. They also used several methods such as the NLTK tool, NRC emotion association lexicon, and rule-based modules to implement their idea to extend Ekman's model of emotions. They achieved good results, and the proposed framework is flexible, and thus can classify emotions in any domain.

Apart from that, numerous researchers have addressed the issue of product or service ranking [69] and recommendation systems [70] in aggregate. In their work, Segev et al. [71] ranked the most influential Instagram users using multiple regression and ranking techniques, including Ridge Regression, Random Forest, and Spearman's rank correlation coefficient. They did not employ a specialized algorithm for ranking. More recent research conducted by Saifullah [72] used the Fuzzy-AHP process to rank tourism trends based on social media data such as TripAdvisor and other online platforms. The highest ranking was for park attractiveness. Other experiments on ranking were performed by Graham and Rodriguez [73]. They examined Reddit's ratings using ranking through textual analysis with keywords to determine users' voting behavior. Cui and Kertesz [74] analyzed the real-time Hot Search List (HSL), which ranks the 50 most popular hashtags depending on the number of search queries on the Sina Weibo website. They found that the COVID-19 outbreak generates new relevant hashtags and showed how they rank in HSL. This approach performed ranking with software HSL rather than an appropriate ranking algorithm.

## 3. Methodology

In light of the current framework, we elaborated on our methodology for both machine learning tasks in order to accomplish our study objectives.

We exclusively used Instagram data to compare the above-mentioned variables to the popularity of artistic memes and associated museum posts. We choose Instagram as a data source because of the diversity of its users' characteristics, including age, gender, education level, interests, personality traits, online behavior and, subsequently, their responses to posts [75]. It thus enables us to acquire a more representative dataset. We selected the number of posts (frequency), the number of likes, the number of comments, and the text emotion as the key indicators of Instagram engagement. Additionally, we picked the number of posts (frequency), likes, and comments to anticipate the most popular memes and artworks. To examine our original research questions (RQs), we employed cutting-edge machine learning approaches.

In this paper, we leverage APIs and machine learning algorithms as methodological tools for empirical evidence. Furthermore, we applied Python programming for the majority of the process. The rest of the methodology is just as follows for both Tasks.

### 3.1. Data Collection
#### 3.1.1. Searching for Data

In the initial step, we conducted a thorough search and compiled a list of eighteen Instagram accounts and ten hashtags dedicated to artistic memes. Then, we searched

for classical art memes that often and consistently occurred on these various accounts. We chose artistic memes that occurred on each account at least ten times. Each time we navigated to a profile, we noted which one appeared the most frequently and then searched for it among the other accounts. We acquired memes from 91 unique works of art between 15 April and 30 April of 2021 (the lockdown period) among approximately 250,000 distinct memes to construct the initial dataset. In the second step, we used Google Images to discover artworks for which we had no title, artist name, or location. Depending on the location of each museum's artwork, we then ran a geographical search on Instagram. From 15 June to 30 June 2021, we were able to discover all of the visitor posts for the 91 paintings by using this method (after the lockdown). We also downloaded all of the discovered museum posts and compiled a second dataset.

### 3.1.2. Downloading the Data

The Instaloader [76] is an open-source Python tool for downloading images or videos, along with their captions and their metadata (e.g., likes, comments, geotags, text, and captions) from Instagram, via a command line interface. This tool enables the user to download from public or private profiles, hashtags, stories, the feed, IGTV, reels, biographic information, and saved media. It downloads the entire post in JSON format along with its images, usually ranging from one to ten. Using the Instaloader Python library version 4.7.4, we gathered $N = 4.526$ ($N = 1.222$ for memes and $N = 3.304$ for museum posts) entire posts from the most popular hashtags and associated geotags with our topic. By setting the resumable_iteration() in our Python code, the hashtag and date of data collection were also specified for museum posts. Since Instaloader enabled us to carry it out, we saved memes to our Instagram collection per artwork and downloaded them as saved posts. Considering Instagram restricts the amount of data that can be downloaded each day, we downloaded the maximum number of posts that Instagram permits. Similarly, we downloaded data for memes and museum posts by continuously shifting the parameters of date, hashtag, or type each time. Python modules such as the Pandas library [77] were also used to convert the JSON strings to CSV. We created two separate datasets, one for artistic memes and the other for museum artwork posts.

### 3.2. Pre-Processing and Transformation

The initial CSV file contained our data in a tabular format with four columns as comments, likes, nwords, and text emotion for memes and museum Instagram posts. The data were in a CSV file which was pre-processed using the open-source software OpenRefine [78]. OpenRefine is a potent tool for dealing privately with messy data, including data cleansing, data, and format transformation. It can manipulate spreadsheets and other file types such as CSV. We imported two CSV datasets described above as input files and produced a processed CSV file for each dataset. According to the GDPR privacy legislation rule, the usernames and IDs were eliminated. Moreover, HTML links, hashtags, and special characters (@, #, &, %, ", +) were removed. Spelling errors (such as "jajajahahaaaja" or "I looooovve this so much" to "I love this so much") and missing values were substituted. Since the bulk of algorithms works with numeric values, the Column Transformer from the SciKit-learn Python library was used to transform categorical values to numeric values when appropriate, such as the text feature [79]. The remaining three columns were formatted numerically.

### 3.3. Emotion Classification

Initially, we also employed the NLTK (Natural Language Toolkit) library, an accessible platform for building Python code using over 50 corpora and lexicons for detecting basic sentiments in textual data [80]. We selected the VADER (Valence Aware Dictionary and sEntiment Reasoner) lexicon [81] because it is well-suited for classifying short texts such as those seen on social media such as Instagram or Twitter as positive, negative, or neutral by using the class SentimentIntensityAnalyzer(). This lexicon identifies emojis as well.

Notably, in our data, the VADER lexicon detected predominantly positive sentiments, with far fewer neutral and less negative sentiments. As a result, we omitted the small proportion of negatives, as this would have generated a substantial imbalance in our data. However, this approach to the scope of our inquiry was really broad. For the classification task, we used a set of positive emotions from a tree-structured list including lust, surprise, and cheerfulness [82,83]. Our concept is substantiated by the most precise definition of emotion, as provided by Thoits in 1990 [84]. He identified four interrelated components of emotion as (1) situational cues, (2) psychological changes, (3) expressive gestures, and (4) an emotion label that names the specific configuration of components. The selected emotions are derived from a broader categorization of emotions as primary, secondary, and tertiary. Since the primary category did not correspond to our idea, we chose the second one. The choice of cheerfulness was dependent on the hilarious tone of the memes' comments. We also selected the emotion of lust since numerous memes deal with sexual subjects, as well as the emotion of surprise, as it indicates the individual's reaction to the meme's content. Using the OpenRefine software [78], we inserted a column labeled emotion as the output column in both datasets and annotated it depending on the textual content sense. The subsequent step involved applying machine learning algorithms.

## 4. Implementation and Results

The objective of the classification task (RQ1) was to establish a link between the four basic features and the emotion variable to comprehend how aesthetic emotions may affect the attractiveness of museums. For spot-checking the classification task, we employed six cutting-edge classification algorithms from the SciKit-Learn library. Then, we picked those that performed better with our data. The LightGBM [85] algorithm was implemented for the ranking test (RQ2, RQ3).

### 4.1. Data Exploration

We began by examining our data to determine its distribution. The initial dataset had 91 paintings by 73 artists, as well as values for the frequency of memes/museum posts, likes, comments, nwords count, and emotion feature. The final two datasets contained simply frequency, comment, like, and nwords values as shown in Table 1. The emotion value was only used for classification.

**Table 1.** Statistical description of Artistic memes dataset and Museum posts dataset.

| Artistic Memes Dataset | | | | | |
|---|---|---|---|---|---|
| Memes Frequency (*N* = 1222) min = 10, max = 34, mean = 12.54, std = 4.25 | | | | | |
| **Features** | ***N*** | **Minimum** | **Maximum** | **Mean** | **Std. Deviation** |
| Comments | 1222 | 40.00 | 40,089.00 | 2808.37 | 6734.39 |
| Likes | 1222 | 902.00 | 756,445.00 | 122,176.18 | 106,819.72 |
| Nwords | 1222 | 0.00 | 627.00 | 134.13 | 103.45 |
| **Museum Posts Dataset** | | | | | |
| Post Frequency (*N* = 3304) min = 4, max = 384, mean = 35.55, std = 42.30 | | | | | |
| **Features** | ***N*** | **Minimum** | **Maximum** | **Mean** | **Std. Deviation** |
| Comments | 3304 | 17.00 | 13,030.00 | 681.46 | 1238.94 |
| Likes | 3304 | 51.00 | 397,639.00 | 24,692.29 | 42,526.43 |
| Nwords | 3304 | 16.00 | 28,461.00 | 1968.46 | 2890.32 |

Museum Posts received fewer likes and comments than memes, since users had fewer followers and interactions than pages of popular memes, which had thousands of followers. However, post frequency was greater in the dataset for museum posts. The same seemed to be accurate for nwords. The number of terms used by visitors to describe their emotions and

art experiences has increased. In addition to that, the memes dataset was predominantly used to explain the memes' content.

*4.2. Classification Task and Findings*

Next, we spot-checked the RQ1 using six cutting-edge supervised learning algorithms from the SciKit-Learn Library [79]. Then, we picked the algorithm that performed better with our data. These algorithms were two linear ones: Linear Regression and Linear Discriminant Analysis, and two non-linear ones: K-Nearest Neighbors and Classification and Regression Trees, Naïve Bayes, and Support Vector Machines.

Using our data, we evaluated the performance of the aforesaid algorithms. Classification and Regression Trees and Support Vector Machines scored better on both datasets than the others.

- **Emotion Identification:** *What emotions inspire people to like, comment, and share posts?*

According to the emotion analysis, three emotions were found as motivating users to comment on and share posts: cheerfulness, lust, and surprise for both datasets containing *N* = 1222 memes and *N* = 3304 paintings, respectively. We discovered that the first dataset contains a greater proportion of lust emotion than the others. In the second dataset, surprise is the most prevalent emotion. This demonstrates, in part, the positive emotions that the memes elicited in their users, as well as they might be surprised when they were discovered in museums.

The goal was to measure how the four features connected to emotion. Moreover, we used the SMOTE (Synthetic Minority Over-sampling Technique) approach from the Imbalanced Learn Library [86], which is fully compatible with the SciKit-Learn API [79] for oversampling imbalanced datasets [87]. By default, SMOTE [87–89] oversamples all minority classes to achieve the same sample size as the majority class. In our scenario, class lust has 511 posts in the first dataset, whereas class surprise has 1306 posts in the second (Table 2). This indicated that the SMOTE method achieved greater precision by balancing the data.

**Table 2.** The frequency of three emotions on the Artistic memes dataset and Museum posts dataset.

| Memes Emotion | N | Frequency | Museum Posts Emotion | N | Frequency |
|---|---|---|---|---|---|
| Cheerfulness | 1222 | 249 | Cheerfulness | 3304 | 997 |
| Lust | 1222 | 511 | Lust | 3304 | 1101 |
| Surprise | 1222 | 462 | Surprise | 3304 | 1306 |
| Valid *N* | 1222 | | Valid *N* | 3304 | |

- *RQ1: How do likes, comments, frequency, and nwords relate to emotion?*

We carried out an experiment on our datasets using the two well-behaved supervised learning algorithms, namely Classification and Regression Trees [90] and Support Vector Machines to examine this link [91], by comparing each painting's likes, comments, and frequency (the total number of occurrences of a painting as a meme or as a museum post) with the expressed emotion. The algorithm, as illustrated in Table 3, achieved more accurate predictions on the emotions of lust and surprise based on the input features we provided. We associated all of the features with each other and classified them based on emotion. For example, we correlated likes and comments with the emotion feature, likes and frequency with the emotion, or each one individually with the emotion.

**Table 3.** Outcomes evaluation by classification report. Correlation between likes, comments, frequency, and emotion.

| Artistic Memes Dataset (*N* = 1222) | | | | |
|---|---|---|---|---|
| **Emotion** | **Precision** | **Recall** | **f1-Score** | **Support** |
| **cheerfulness** | 0.86 | 1.00 | 0.92 | 6.00 |
| **lust** | 1.00 | 1.00 | 1.00 | 4.00 |
| **surprise** | 1.00 | 0.88 | 0.93 | 8.00 |
| **accuracy** | | | 0.94 | 18.00 |
| **macro avg** | 0.95 | 0.96 | 0.95 | 18.00 |
| **weighted avg** | 0.95 | 0.94 | 0.94 | 18.00 |
| Museum Posts Dataset (*N* = 3304) | | | | |
| **Emotion** | **Precision** | **Recall** | **f1-Score** | **Support** |
| **cheerfulness** | 0.67 | 0.67 | 0.67 | 6.00 |
| **lust** | 0.78 | 0.78 | 0.78 | 9.00 |
| **surprise** | 1.00 | 1.00 | 1.00 | 5.00 |
| **accuracy** | | | 0.80 | 20.00 |
| **macro avg** | 0.81 | 0.81 | 0.81 | 20.00 |
| **weighted avg** | 0.80 | 0.80 | 0.80 | 20.00 |

Specifically, the algorithms effectively approached our data in all correlations up to 81–96% or higher in some cases. The standard deviation was as follows; Classification and Regression Trees (0.07%) and Support Vector Machines (0.10%) for the memes dataset and Classification and Regression Trees (0.06%) and Support Vector Machines (0.08%) for the museum posts dataset. It was crucial to select the appropriate metric for model evaluation. The accuracy metric is inapplicable as a performance indicator for imbalanced classification datasets. Even though we used the SMOTE method, the test yielded slightly lower percentages than anticipated. To overcome this, we employed a set of well-known metrics for both datasets. The classification report offered a breakdown of each class (cheerfulness, lust, and surprise) by precision, recall, f1-score, and support metrics [92], achieving great results. Based on these favorable results, we concluded that our classification model was quite accurate. In addition, we set up a 10-fold cross-validation and 0.25% validation test for all iterations in both datasets. By using the resampling technique, each training dataset sample was utilized k to 10 times to train the model. As a result, all samples were evaluated to understand the accuracy of the model. For the two research questions in which we applied supervised machine learning algorithms, we followed the same procedure. The model has successfully classified new instances of artistic memes and museum posts, respectively.

To have a deeper insight into our data, we split the features so that we could run two experiments with distinct associations.

Using the two well-behaved algorithms, Classification and Regression Trees and Support Vector Machines, we performed the same experiment on our datasets to investigate this relationship by comparing each painting's likes, comments, and nwords (total number of words) with the conveyed emotion. The algorithm, as shown in Table 4, also achieved more accurate predictions on the emotions of lust and surprise based on the input features we provided. Likewise, in this case, we correlated all features with each other and classified them according to the expressed emotion. We associated, for example, likes and comments with the emotion feature, likes and nwords (total number of words) with emotion, or each independently with emotion. The fact that the second dataset had a greater number of nwords implies that museum visitors described their art experiences. We assumed that when viewers responded favorably to artistic memes, they would eventually be surprised when they encountered specific artworks in museums. This is also evidenced by the aesthetic experience paradigm [59], which asserts that prior encounters with art contribute to the final judgment and aesthetic experience.

**Table 4.** Outcome evaluation by classification report. Correlation between likes, comments, nwords, and emotion.

| Artistic Memes Dataset (*N* = 1222) | | | | |
|---|---|---|---|---|
| **Emotion** | **Precision** | **Recall** | **f1-Score** | **Support** |
| **cheerfulness** | 0.86 | 1.00 | 0.92 | 6 |
| **lust** | 1.00 | 1.00 | 1.00 | 4 |
| **surprise** | 1.00 | 0.88 | 0.93 | 8 |
| **accuracy** | | | 0.94 | 18 |
| **macro avg** | 0.95 | 0.96 | 0.95 | 18 |
| **weighted avg** | 0.95 | 0.94 | 0.94 | 18 |
| Museum Posts Dataset (*N* = 3304) | | | | |
| **Emotion** | **Precision** | **Recall** | **f1-Score** | **Support** |
| **cheerfulness** | 0.75 | 1.00 | 0.86 | 6 |
| **lust** | 1.00 | 0.78 | 0.88 | 9 |
| **surprise** | 1.00 | 1.00 | 1.00 | 5 |
| **accuracy** | | | 0.90 | 20 |
| **macro avg** | 0.92 | 0.93 | 0.91 | 20 |
| **weighted avg** | 0.93 | 0.90 | 0.90 | 20 |

Similarly, for the following relationship, the algorithms effectively approached our data at a rate of up to 75–96% or higher in some cases. The standard deviation was the same as previously; Classification and Regression Trees (0.07%) and Support Vector Machines (0.10%) for the memes dataset and Classification and Regression Trees (0.06%) and Support Vector Machines (0.08%) for the museum posts dataset. In particular, we established a 10-fold cross-validation and 0.25% validation test for all iterations in both datasets. We concluded, based on these findings, that our classification model was quite accurate. New examples of artistic memes and museum posts were successfully classified by the algorithm. In RQ1, the major category in the dataset of memes was lust, while the major category in the dataset of museum posts was surprise. This demonstrated that the visitors were too mesmerized by the paintings they saw as memes and shared their experiences.

Despite that, other researchers have also explored the popularity of art on Instagram, albeit not from an emotion-based meme perspective, as we conducted in this study. For example, Kang, Chen, and Kang [53] investigated the most-liked artworks and the relationship between Instagram interactions and the creative artistic process. Visitors to museums commonly utilize mobile devices to record their impressions. Budge and Burness [21] concentrated their Instagram posts with specific hashtags and geotags on photographic practices and playfulness rather than art sharing or popularity. Other studies focusing primarily on the popularity of Instagram posts, but not in the context of art, such as the emoji-based research by Mazloom et al. [93], have been undertaken. Others focused on extracting emotions from images as opposed to words [94], while others [95] simply considered the descriptive power of hashtags.

### 4.3. Ranking Task and Findings

The ranking is a fundamental problem in machine learning in which a list of items is ranked according to their relevance to a certain task. It has multiple uses, including e-commerce [96], recommendation systems [97], webpage [98] or product rating [99], query auto-completion [100], and image search [101], among others. In addition, learning-to-rank models attempt to predict the rank (or relative order) of a list of items for a particular ranking challenge [102]. The algorithm Light Gradient Boosting Machine [85] was applied for the Ranking Task (RQ2, RQ3). The LightGBM framework by Microsoft combines tree-based learning algorithms (ensemble model) that are sequentially trained. The LightGBM's ability to perform regression, classification, and ranking with learning-to-rank functions

is one of its coolest properties [103]. We decided to employ a ranking algorithm since it is the only method for determining which memes and artworks are more popular than others. This approach also reinforces our attempts to appreciate the impact of artworks and previous research questions; however, from a novel perspective.

### 4.3.1. Data Handling

The implementation of the ranking involves the formulation of two specific datasets. The LightGBM algorithm [85] works with numerals. Moreover, the data should be sorted into groups beforehand. We prepared two datasets of 91 artworks, respectively (the memes dataset $N = 1222$ and the museum posts dataset $N = 3304$), having six columns with similar features as ID, label, comments, likes, memes frequency, and museum post frequency. The "label" column refers to the relevance of features, as assessed by their impact on what is popular or less popular. Some researchers employed user-level metrics such as the number of followers, also by quantifying reputation, bow-tie group, inactivity, and social influence to evaluate popularity [104]. In their experiments about music popularity prediction, Lee and Lee [105] used multiple popularity metrics extracted from song rankings which were measured using rank scores such as max rank (the maximum rank score of a song during the whole specific period) and rank (the rank of the song). They assumed, for example, that the most popular song on a chart of 100 would have a rank of 100, whereas the least popular song would have a rank of 1. Chapelle and Chang [106] also employed in their machine learning task the labels {0,1,2,3,4} for document retrieval ranking, where the more positive number denoted higher relevance and the more negative lower relevance.

Similarly, in this study, we set the values for this feature to {0,1,2,3}, where 0 indicates the absolute negative correlation, 1 represents the least relevance, 2 represents the most relevance, and 3 indicates the absolute positive correlation. Within the two datasets, each painting was represented as a distinct group. We assigned each sample across each group a value between 0 and 3 depending on the number of interactions achieved per post. For posts with the most interactions, we assigned the value 3. For posts with the fewest interactions, we assigned the value 0. Due to its relevance, the remaining medium values were assigned the values 1 and 2, respectively. All samples were carefully assigned using the OpenRefine software.

### 4.3.2. Implementation Outline

In general, Light GBM is a rapid training speed algorithm with higher efficiency, reduced memory consumption, enhanced accuracy, and the ability to handle varying data scales. For this reason, we selected the LightGBM Ranker [105] to implement the following queries. As an objective function for ranking, we used LambdaRank. The optimization of ranking functions, such as nDCG (Normalized Discounted Cumulative Gain), which is also used in our code, has proven to be highly effective by LambdaRank. The nDCG [107] is a popular ranking metric that evaluates the gain of a sample based on its position. More specifically, a relevant sample placed at the top of predictions has a greater gain than one placed at the bottom of the predicted data, which has a lower gain. As previously described, the LightGBM algorithm can perform classification, regression, and ranking. Combining relevance, importance, and content score, ranking algorithms are used to provide query results based on the user's preferences. It can handle two ranking approaches such as pairwise and listwise ranking. In the pairwise approach, each pair of instances is compared against the test set to determine the optimal ordering for that pair. The objective of the ranker is to reduce the number of inversions in ranking, while in the Listwise approach, it is to examine the whole list of samples and use sub-techniques such as the nDCG measure to determine the optimal ordering [108]. As in our scenario, we desired to export the artworks in descending score order based on their similarity, beginning with the highest and finishing with the lowest. Then, let us clarify below how our data behaved with the algorithm.

- *RQ2: What is the popularity of artistic memes?*

Taking advantage of the structure of our data, we performed a ranking experiment to identify the most popular classic art memes among many others. We opted for a normal sample size ($N = 1222$) due to Instagram's daily rate limitations for data downloading. The dataset comprised six columns: ID, label (relevance score 0–3), comments (count of comments per post), likes (count of likes per post), meme frequency (times of occurrence per post as a meme), and museum post frequency (times of occurrence per meme as a museum visitor caption). We included the occurrence frequency of memes and museum posts in both datasets as we desired a more complete outcome by considering all of the available parameters. Moreover, we designated approximately 81% (72 groups) for training and the remaining 19% (19 groups) for validation testing in both datasets, instead of the classic 80% for training and 20% for validation because we did not wish to separate the posts of the same group. Afterward, we picked 10% of the data from the validation test set with the intent of identifying the groups to which these memes belong. Then, we selected the unique values in this set, where each unique value indicated a distinct group. We observed that taking 10% of the posts led to seven memes being the most popular, while taking 20% of the posts led to nine memes being the most popular. In other words, increasing the percentage by 10% only brought two memes (Portrait of Dorothea Schlegel by Anton Graff and Miniature Self Portrait by Louis Marie Autissier) to the list of the most popular (Table 5).

**Table 5.** The top 7 of the most popular classical art memes by ranking outcomes.

| The Final Rank of the Most Popular Classical Art Memes ($N = 91$) | | | |
|---|---|---|---|
| **Classical Art Meme** | **Artist** | **Group id** | **Predicted Ranking Score** |
| 1. The Soul of the Rose | John William Waterhouse | 39 | 8.57 |
| 2. The Bookworm | Carl Spitzweg | 62 | 8.23 |
| 3. The Blessing Christ | Jean Auguste Dominique Ingres | 1 | 8.15 |
| 4. Mars & Venus, Allegory of Peace | Louis Jean François Lagrenée | 97 | 7.98 |
| 5. Portrait of Cardinal Pietro Bembo | Tiziano Vecelli | 74 | 7.98 |
| 6. Portrait of a Man | Hans Memling | 101 | 7.74 |
| 7. La Pensée (The Thought) | Jean Despujols | 105 | 7.70 |

According to the ranking task's overall measurement data, the score of the most popular classical art memes ranges from 8.57 (the higher-ranking value) to 6.95 (the lowest ranking value). Taking the top 10% of the validation-ranked data revealed that seven classical art memes (see Table 5) were the most popular. Likewise, taking the top 20% of the validation-ranked data revealed that only one meme was common with the second dataset's museum posts, and this was La Pensée (The Thought) by Jean Despujols. However, this did not impair our idea. Given that both datasets contained similar artworks, it enhanced both our research questions and the reliability of the model.

- *RQ3: What is the popularity of museum artworks?*

Similarly, we conducted a ranking experiment to determine the most popular museum-based paintings encountered as an artistic meme. We selected a normal sample size ($N = 3304$) due to Instagram's daily data download rate limits. Likewise, the dataset included the six columns as previously: ID, label (relevance score 0–3), comments (count of comments per post), likes (count of likes per post), meme frequency (occurrences per post as a meme), and museum post frequency (times of occurrence per meme as a museum visitor caption). We allocated approximately 79% (72 groups) to training and 21% (19 groups) to validation testing. Then, we selected 10% of the data from the validation test set to identify the groups to which these artworks belong. Afterward, we selected the unique values from this set, where each unique value represented a separate group. Moreover, in this case, we noticed that taking 10% of the posts yielded 12 of the most popular artworks, while taking 20% of the posts yielded 15 of the most popular artworks. In another sense, increasing

the proportion by 10% led to the expansion of only three artworks (Miniature Self Portrait by Louis Marie Autissier, La Pensée (The Thought) by Jean Despujols, and Portrait of a Man by Hans Memling) to the list of the most popular museum artworks (Table 6). It is important to point out that four classical art memes were common with the meme posts from the first dataset. These memes were: The Soul of the Rose (John William Waterhouse), The Bookworm (Carl Spitzweg), The Blessing Christ (Jean Auguste Dominique Ingres), and Mars and Venus, Allegory of Peace (Louis Jean François Lagrenée). As illustrated below, the tie of some results emerged since it was the absolute highest ranking that the algorithm could assign to these artworks, and predicted ranking scores are quantized. In a broader sense, the overall ranking does not necessarily imply that the artwork is the most popular; it simply indicates the ranking. In addition, we are not interested in the most popular museum post for this question, but rather the most popular artworks.

**Table 6.** The top 12 most popular museum artworks by ranking outcomes.

| | The Final Rank of the Most Popular Museum Visitor's Posts (*N* = 91) | | | |
|---|---|---|---|---|
| | **Artwork** | **Artist** | **Group id** | **Predicted Ranking Score** |
| 1. | Charles I in Three Positions | Anthony van Dyck | 9 | 10.05 |
| 2. | The Blessing Christ | Jean Auguste Dominique Ingres | 1 | 10.05 |
| 3. | The Soul of the Rose | John William Waterhouse | 39 | 10.05 |
| 4. | The Love Potion | Evelyn de Morgan | 96 | 10.00 |
| 5. | The Bookworm | Carl Spitzweg | 62 | 9.72 |
| 6. | Travesuras de la Modelo | Raimundo de Madrazo y Garreta | 103 | 9.72 |
| 7. | Carolus Duran | John Singer Sargent | 106 | 9.69 |
| 8. | The Proposition | Arturo Ricci | 107 | 9.69 |
| 9. | The Unconditional Lover | Vittorio Reggianini | 94 | 9.69 |
| 10. | A Girl with a Dead Canary | Jean Baptiste Greuze | 102 | 9.69 |
| 11. | Mars & Venus, Allegory of Peace | Louis Jean François Lagrenée | 97 | 9.69 |
| 12. | Gathering Flowers | Albert Lynch | 95 | 9.62 |

The results thus obtained are compatible with the previous measurements. Over this, it is not deemed necessary to repeat the classical art memes that are present as common in the initial dataset. A similar approach is used for the museum posts ranking task. According to the ranking task's overall measurement data, the order of the most popular museum artworks which were most photographed approximately ranges from 10.05 (the higher-ranking value) to 9.06 (the lowest ranking value). Interestingly, the association between memes and common museum posts emphasizes the validity of our model. We observed that popular memes were frequently photographed as museum artworks and shared on Instagram accounts. At this point, we refer to the five common artworks discovered in both datasets. We additionally noticed artworks that were popular as memes but not as museum posts, as well as artworks that were extremely popular as museum posts but not as memes. Moreover, it is important to stress that the ranking scores in the museum posts dataset (Table 6) were significantly higher than the ones in the memes posts. Our experiments corroborated our assumptions that memes can influence the aesthetic experience and photographic practices in museums and their social media presence.

It is interesting to observe that the predicted ranking score distributions of the four common groupings across memes and museum posts are extremely similar. This is illustrated in Figures 2 and 3. The vertical axis shows the score post, while the horizontal axis represents the posts. In both cases, it seems that our ranking model performed extremely well and accurately in respect to ranking memes and artworks. Furthermore, the smooth curves of the common artworks in both memes and museum posts graphs show a possible interaction. In addition, this may contribute to the overall artwork's popularity and museum attractiveness.

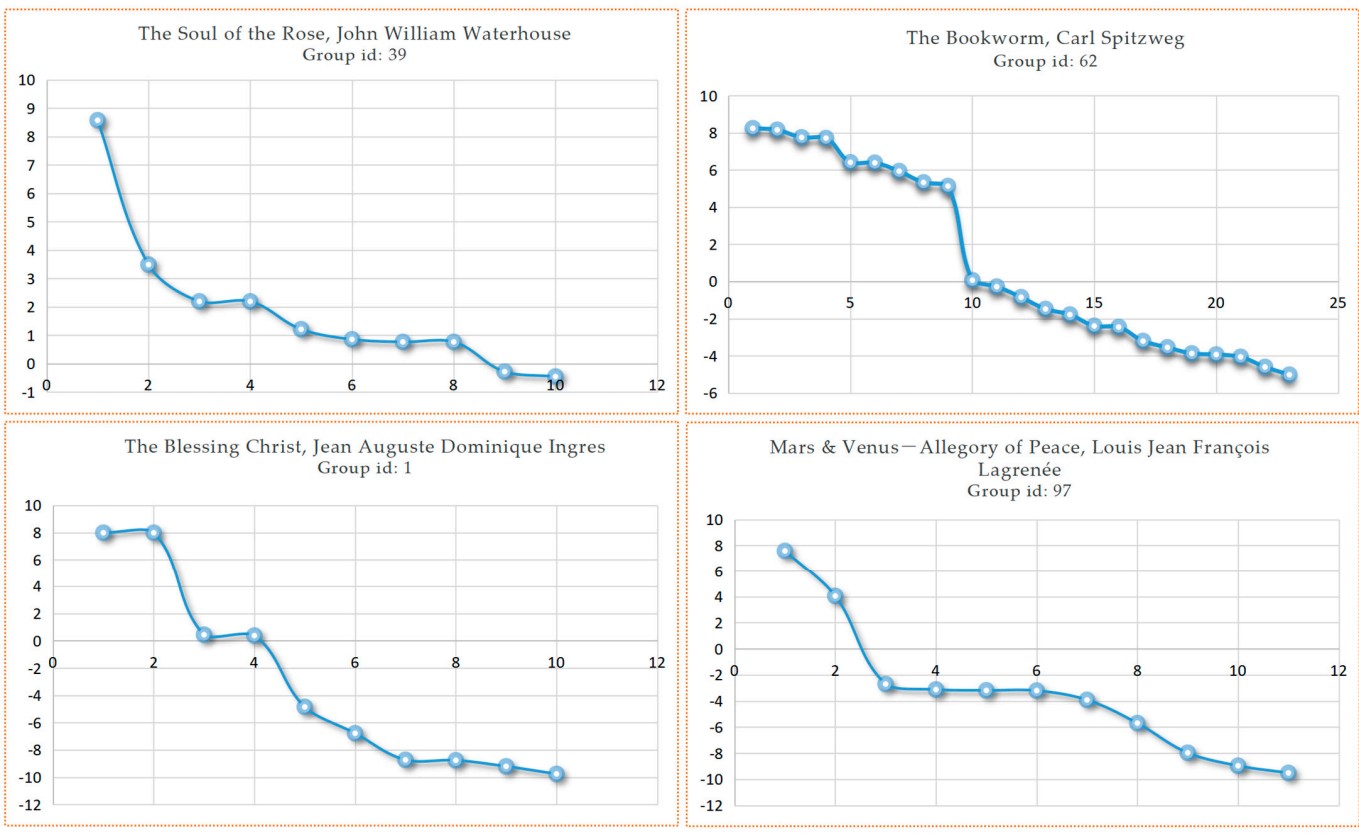

**Figure 2.** Distribution of the predicted ranking score for the 4 common artworks (memes).

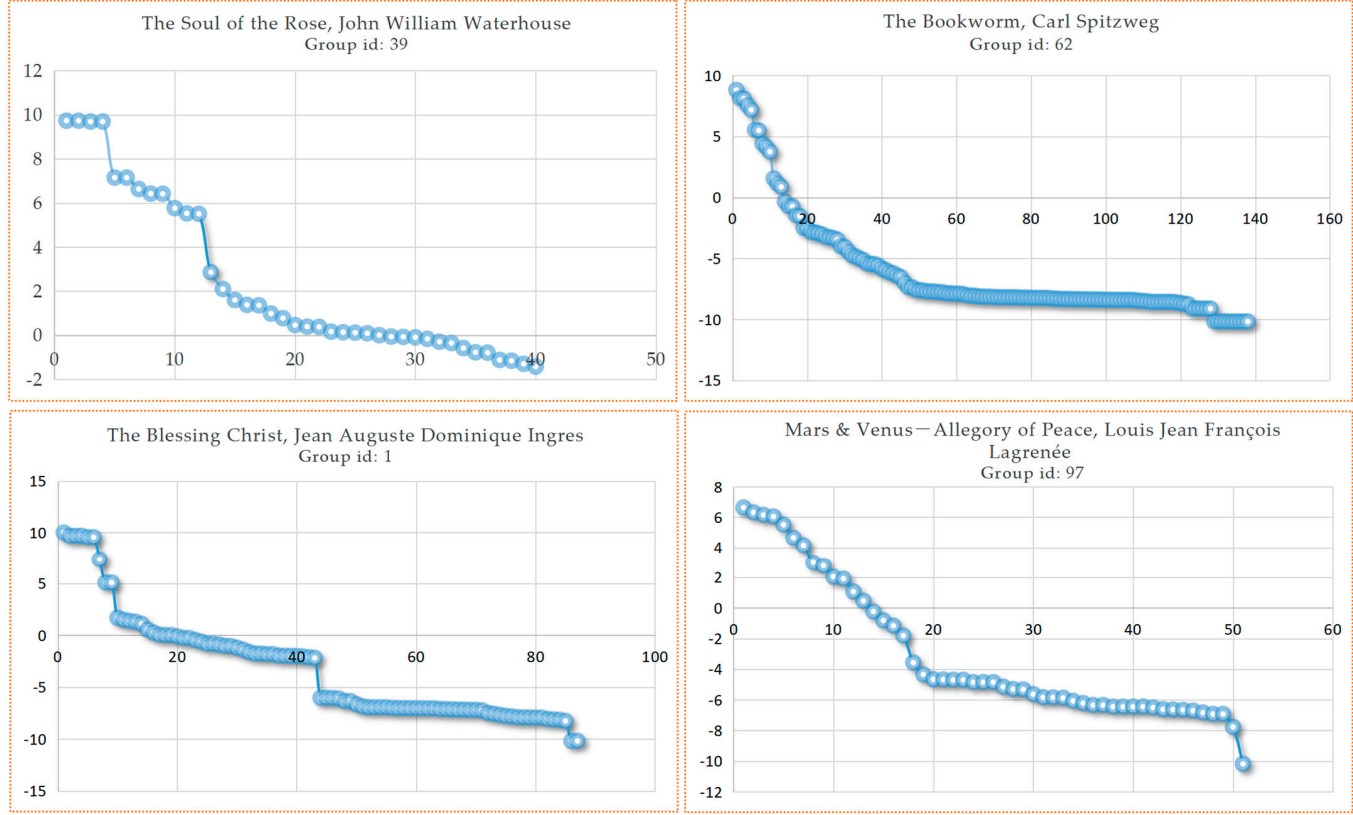

**Figure 3.** Distribution of the predicted ranking score for the 4 common artworks (museum posts).

## 5. Limitations

This was the first attempt to examine artistic memes regarding museum and art popularity by using classification and ranking algorithms. To address this gap thoroughly, we selected a normal sample of data, but we expected to elaborate on it in future papers. The most difficult aspect of the first step was establishing the frequency of meme appearances across all Instagram accounts. We chose Instagram as the main source of data above other social media platforms because it is the one where memes transcend and are shared rapidly and extensively. Our research was not user-centric but rather reaction-centric. There was no analysis of gender, age, country of origin, degree of education, or interests. We did not inspect any artwork attributes or the museum space.

It is necessary to describe the technical difficulties we encountered at this point. Downloading the posts with the Instaloader library posed the second and most essential obstacle. The implementation of changes to their API and a harsh rate limit took a very long time. It was necessary to highlight that downloading 150 to 200 posts takes around three hours. Since bulk data downloads are prohibited, the operation was carried out in stages. Instagram imposes a rate limit. We had to create multiple Instagram accounts to obtain the data because of Instagram limitations.

We selected 91 artistic memes out of the approximately 250,000 artistic memes daily monitored, since only these were repeated ten or more times in each account. Among the millions of museum posts, it was virtually hard to discover individually all of the related artworks in museum geotags that we encountered as artistic memes. Nevertheless, we were mainly successful in discovering them. In addition, we endeavored to explore more deeply into the art popularity issue by setting more complex research questions, collecting more data, and including variables such as aesthetic experience, artwork or space parameters, and social media users' reactions.

## 6. Discussion

This paper contributes to the ongoing discussions about the aesthetic experience and the use of social media such as Instagram in museums. To clarify various associations between emotions, museum visitors' posts, and memes, we conducted machine learning experiments such as supervised and ranking tests. The fact that positive emotions predominated in the preceding scenarios exemplifies that both museum visitors and Instagram users enjoy interacting with artworks in the form of paintings or memes. We were able to perform the ranking and identify the most popular artworks based on the criteria we established in the methodology. This was accomplished due to the large number of artworks and memes shared on Instagram, as well as the high proportion of user interactions, for example, the total number of likes, comments, and so on.

The originality of our concept can be found in the manner in which the contextual components defined in our research questions were linked. We presented a novel approach to measuring aesthetic experience using the most popular artworks and memes. Given that the majority of studies on this topic are theoretical in nature, our findings differ substantially from previous results reported in the literature. In his recent paper, Nieubuurt [109] addressed the idea of Internet memes as leaflet propaganda of the digital era. He remarked on Internet memes' ephemeral and controversial content, as well as their ability to maintain or rupture social bonds. Moreover, Wiggins [110], in his book *The Discursive Power of Memes in Digital Culture*, extensively studied the role of memes in a cultural, economic, and political context. Wagener [111] investigated the social impact of memes and GIFs from a post-digital perspective. He explored their communicative role, recognizing how they can represent a new form of human discourse, culture, and collaborative creativity. Our supervised learning experiments supported the hypothesis by proving a positive link between the test variables. In addition, the most popular artworks and memes received a decent rating in the ranking tests. According to the research findings, visitors and Instagram users interacted with the art and had aesthetic experiences through a physical or digital presence. We were aware of the technical and scientific limitations of our research, which

are discussed above. The difficulty in collecting data from many museum hashtags, geotags, and other public accounts associated with memes and artworks evidenced these constraints. Our findings appeared to be well-justified and addressed the research questions posed at the beginning of the paper. Nonetheless, we want to convey that either the meme concept or the popularity of museum posts is multidimensional. This could be probed in a variety of contexts and with a diverse set of variables each time. In order to publish in the future, we have already begun researching other aspects of the aesthetic experience and the popularity of artworks.

## 7. Conclusions and Further Research

The purpose of the foregoing outline was to find out the nexus between uploaded museum visitors' posts and their emotions, as well as the popularity of artworks and memes using cutting-edge methods. Despite the limitations described above, we may conclude, based on our investigation's findings, that these outcomes provide some conclusions on the topic, which will be examined further in future studies. We conducted an exhaustive analysis using several supervised machine learning tasks and ranking tasks [112] to address the research questions.

The pervasive use of social networking in many aspects of life has influenced our appreciation for the arts and culture. Hence, we chose the most popular classical art memes because they are an eye-catching social media trend that often goes viral. According to our findings, people may have been surprised to encounter artworks that they had previously viewed as memes during both scheduled and spontaneous visits. Based on the outcomes of the classification experiments, we discovered that lust and surprise were the most prominent emotions across both datasets. In addition to that, the ranking outcomes validated our premise about what types of popular memes and artworks are shared in a certain period across multiple accounts.

This study provided additional evidence that the digital age has had a significant impact on the aesthetic experience of museums. A large number of Instagram influencers are photographed daily in iconic cities, monuments, and museums to advertise well-known brands or services. For instance, the travel, cuisine, and fashion bloggers and trendsetters we regularly encounter promote lavish suites or various hotel services, restaurants and cafés, apparel, cosmetics, and other commodities [14]. As Lev Manovich 115points out, this is what the notion of Instagrammism comprises [113].

In conclusion, we pursued to gain future insight into the art popularity discussion by posing more sophisticated research questions, obtaining more artworks, and integrating aesthetic experience (emotional responses and personal qualities), artwork features (form, colors, material), or spatial factors (space design, exhibition practices), as well as social media user interactions. We hope this research will aid future social media studies on museum art appreciation and appeal and that our findings will encourage further academic research in sectors such as art curation, cultural and heritage management, advertising, aesthetics, fine arts, and culture analytics.

**Author Contributions:** Conceptualization, S.V. and M.P.; formal analysis, S.V.; investigation, S.V.; methodology, S.V. and M.P.; software, S.V. and M.P.; resources, S.V.; data curation, S.V. and M.P.; writing—original draft, S.V.; visualization, S.V.; supervision, M.P.; writing—review and editing, S.V. and M.P. All authors have read and agreed to the published version of the manuscript.

**Funding:** This research was funded by the project HAL (Hub of Art Laboratories) *MIS:5047267* code 80504, ESPA 2014–2020, EPAnEK, co-financed by Greece and the European Union—European Regional Development Fund and implemented at the Ionian University, Corfu. The APC was also funded by HAL (Hub of Art Laboratories) https://hal.avarts.ionio.gr/en/ (accessed on 1 August 2022).

**Data Availability Statement:** The data presented in this study are openly available in Zenodo at 10.5281/zenodo.7121364 (accessed on 1 August 2022).

**Conflicts of Interest:** The authors declare no conflict of interest.

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
