# Peer review of "An Examination of Classical Art Impact and Popularity through Social Media Emotion Analysis of Art Memes and Museum Posts"

_information, doi:10.3390/info13100468_

Round 1
Reviewer 1 Report
The authors show a very interesting study using social media and memes to promote classical art in museums.
I believe that the authors should expand the Introduction more. This is my opinion, although this is not a critical requirement.
The well -built and described chapter "Related Works" deserves recognition. Well written, shows that the authors have well received the topic.
In my opinion, the methodology described satifiesively.
Note, partly technical. The same precision should be used in the tables with the same size. For example, in Table 1: Min: 40 Max: 40089 BUT MEAN: 2508.37 It should be in X.XX format everywhere.
Similarly in Table 3. This time the accuracy, however, X.XXX at the frequency.
I like that the authors not only present the results, but also comment on them, achieving the resulting effect are the well -known works of other teams.
It's good that the authors show limits. This is important.
However, I lack a separate chapter Discussion.
After corrections and supplementation with the chapter, I think that it is worth publishing.
Author Response
Answer to reviewer 1
The authors show a very interesting study using social media and memes to promote classical art in museums.
I believe that the authors should expand the Introduction more. This is my opinion, although this is not a critical requirement.
We have added to the introduction section (Lines 60 – 72 of the revised version of the paper) the editor’s recommendation about what a visitor can find in a museum. In addition, we added lines 33 – 37 and lines 50 – 55 to enrich the aesthetic experience part of the introduction section.
The well -built and described chapter "Related Works" deserves recognition. Well written, shows that the authors have well received the topic.
In my opinion, the methodology described satifiesively.
Note, partly technical. The same precision should be used in the tables with the same size. For example, in Table 1: Min: 40 Max: 40089 BUT MEAN: 2508.37 It should be in X.XX format everywhere.
Similarly in Table 3. This time the accuracy, however, X.XXX at the frequency.
We have changed table 1 and table 3 so that all numbers have the same precision.
I like that the authors not only present the results, but also comment on them, achieving the resulting effect are the well -known works of other teams.
It's good that the authors show limits. This is important.
However, I lack a separate chapter Discussion.
We have added a discussion part (Section 6): Lines 761 – 797.
After corrections and supplementation with the chapter, I think that it is worth publishing.
We want to thank the reviewer for the kind comments.
Reviewer 2 Report
1. The subject matter is interesting.
2. The article is based on an impressive literature review.
3. I have doubts whether tables should be signed over. I would also not divide them between the parties, because it detracts from their content.
4. There is no clearly separated discussion part.
5. The aim of the work and research questions have been clearly formulated.
Author Response
Answer to reviewer 2
- The subject matter is interesting.
We would like to thank the reviewer for this comment.
- The article is based on an impressive literature review.
Thank you for the kind comment.
- I have doubts whether tables should be signed over. I would also not divide them between the parties, because it detracts from their content.
All tables are formatted according to MDPI’s guidelines with no sign over. We merged the divided table 4.
- There is no clearly separated discussion part.
We have added a discussion part (Section 6): Lines 761 – 797.
- The aim of the work and research questions have been clearly formulated.
We once again want to thank the reviewer for the nice comments.
Round 2
Reviewer 1 Report
The authors explained my doubts and corrected the indicated places.